# Autoclaving Nest-Material Remains Influences the Probability of Ectoparasitism of Nestling Hoopoes (*Upupa epops*)

**DOI:** 10.3390/biology9100306

**Published:** 2020-09-23

**Authors:** Mónica Mazorra-Alonso, Manuel Martín-Vivaldi, Juan Manuel Peralta-Sánchez, Juan José Soler

**Affiliations:** 1Departamento de Ecología Funcional y Evolutiva, Estación Experimental de Zonas Áridas del Consejo Superior de Investigaciones Científicas, 04120 Almería, Spain; 2Unidad Asociada Consejo Superior de Investigaciones Científicas: Coevolución: Cucos, Hospedadores y Bacterias Simbiontes, Universidad de Granada, 18071 Granada, Spain; mmv@ugr.es; 3Departamento de Zoología, Universidad de Granada, 18071 Granada, Spain; 4Departamento de Microbiología, Universidad de Granada, 18071 Granada, Spain; jmps@ugr.es

**Keywords:** bacterial community, chemical volatiles, ectoparasitism, hoopoe, nest-material, uropygial gland

## Abstract

**Simple Summary:**

Microorganisms may have direct negative effects on their animal hosts and cause diseases, but some others provide animals with protection against infections, parasites, and even predators. However, parasites or predators might take their cue from odors produced by bacteria, even those from protecting microorganisms, which could turn net benefits of bacteria that are a priori considered beneficial into a neutral or even negative outcome. This possibility has scarcely been studied in wildlife populations; we manipulated the bacterial community of nest-material of hoopoes and detected a negative effect in terms of the intensity of parasitism by blood-sucking flies that nestlings suffer. We also detected a positive link between the bacterial density of the nest-materials and the intensity of ectoparasitism, which further points at the importance of the bacteria determining the level of parasitism. Blood-sucking ectoparasites are also disease vectors, affecting both humans and livestock, and considering the role of the bacterial environment might help to establish new transmission control protocols.

**Abstract:**

Nest bacterial environment influences avian reproduction directly because it might include pathogenic- or antibiotic-producing bacteria or indirectly because predators or ectoparasites can use volatile compounds from nest bacterial metabolism to detect nests of their avian hosts. Hoopoes (*Upupa epops*) do not build nests. They rather reuse holes or nest-boxes that contain remains of nest-materials from previous breeding seasons. Interestingly, it has been recently described that the nest’s bacterial environment partly affects the uropygial gland microbiota of hoopoe females and eggshells. Blood-sucking ectoparasites use chemical cues to find host nests, so we experimentally tested the hypothetical effects of microorganisms inhabiting nest-material remains before reproduction regarding the intensity of ectoparasitism suffered by 8-day-old nestling hoopoes. In accordance with the hypothesis, nestlings hatched in nest-boxes with autoclaved nest-material remains from the previous reproductive seasons suffered less from ectoparasites than those hatched in the control nest-boxes with nonautoclaved nest-material. Moreover, we found a positive association between the bacterial density of nest-material during the nestling phase and ectoparasitism intensity that was only apparent in nest-boxes with autoclaved nest-material. However, contrary to our expectations, nest bacterial load was positively associated with fledgling success. These results suggest a link between the community of microorganisms of nest-material remains and the intensity of ectoparasitism, and, on the other hand, that the nest bacterial environment during reproduction is related to fledging success. Here, we discuss possible mechanisms explaining the experimental and correlative results, including the possibility that the experimental autoclaving of nest material affected the microbiota of females and nestlings’ secretion and/or nest volatiles that attracted ectoparasites, therefore indirectly affecting both the nest bacterial environment at the nestling stage and fledging success.

## 1. Introduction

Microorganisms are a selective force driving the evolution of macroorganisms at different levels. Beyond their well-documented deleterious effects as infectious agents, bacteria are considered the second genome of host organisms because they are essential for their correct development and functioning [1]. The protection against pathogens by stimulating the immunological system or directly synthesizing antimicrobial peptides (e.g., bacteriocins) [2], as well as their role producing essential nutrients for their hosts [3,4], are some classic examples of the beneficial effects of bacteria to their hosts. Interestingly, the role of bacteria in the modulation of complex animal behaviors has gained importance [5,6]. For instance, bacteria are responsible for some volatiles signaling host characteristics [6,7,8] and, thus, might play an important role in the evolution of social communication. Bacteria metabolisms provide their hosts with an odor signature that broadcast information about the sex, social status, kinship, and even group membership of the individuals [9,10]. Odors for bacterial metabolism might also attract predators and/or parasites with a good olfactory system [11,12,13,14,15]. We know, for instance, that two species of mosquitoes, *Anopheles gambiae* and *Anopheles arabiensis,* have a greater attraction for traps that contains bacterial communities of its natural hosts, humans and birds, respectively [16]. Moreover, *Anopheles gambiae* are even able to distinguish particularities of individual humans depending on skin bacterial richness and diversity through cueing on the volatiles that different bacterial communities produce [14,15,17,18]. Evidence of the expected association between symbiotic bacteria of hosts and the probability of parasitism, however, comes from laboratory experiments and human parasites. Studies on natural conditions are therefore necessary to confirm the hypothetical role of bacterial communities of animals in producing volatile compounds that parasites use to detect hosts.

Physiological activity of nestlings and parents, as well as relatively appropriate climatic conditions of avian nests for bacterial growth (particularly those built in holes) [19], make the avian nest environment an ideal niche for testing the hypothetical role of symbiotic bacteria determining the probability of parasitism. Together with incubation and brooding activity, which generate the optimal temperature [20], the accumulation of organic materials in nests, including feces, feathers, skin, and remains of food, would enhance bacterial growth in these environments. Ectoparasites or predators might then take their cue from particular volatiles from the metabolism of bacteria growing in avian nests to detect active hole-nests. In accordance with this hypothesis, Azcárate-Gracía and coauthors detected that the breakage of fecal sacs increased the bacterial density of spotless starlings (*Sturnus unicolor*) nests and that nestlings in experimental nests tended to suffer more from ectoparasites and experienced a higher depredation rate than those in control nests [21]. However, they did not manipulate the bacterial density of nests directly but added broken fecal sacs to nests of a species in which parents remove nestlings’ feces. Although the performed experiment affected nests’ bacterial community, a more direct experimental modification of the bacterial nest environment is necessary to check the effects of nest bacterial community on parasitism and predation.

We here performed such an experiment in artificial hole-nests used by hoopoes *Upupa epops*. The hoopoe is a particularly relevant biological model for testing the hypothesis that ectoparasites take their cue from volatiles from symbiotic bacteria of avian nests. Preen glands of breeding females and nestlings host mutualistic bacteria that produce antibiotic substances [22,23,24,25,26] and a variety of malodorous volatiles [26] that some ectoparasites might use to detect active nests. Furthermore, it has been experimentally demonstrated that nest-material remains from previous hoopoe reproductive events affect the microbial community of the uropygial secretion of adults and nestlings [27]. Thus, a link between nest bacterial communities and volatiles from symbiotic bacteria likely exists in this species. The expected finding may point out that nest bacterial community and their metabolism would be responsible for volatiles that ectoparasites might track to detect active hoopoe nests. We evaluated the intensity of ectoparasitation by *Carnus hemapterus* flies that parasitize nestlings and incubating adults of several birds species [28,29,30,31]. The life cycle of this ectoparasite fly is synchronized with that of its hosts [32]. In spring, winged adults emerge from overwintering pupae inside avian nest-materials from previous reproductions. Adult flies stay in the same cavity or disperse, looking for holes with avian breeding activity [33]. The peak of *C. hemapterus* abundance occurs before the start of nestling feathering, which, in hoopoes, occurs 6–8 days after hatching, and decreases thereafter [34]. Once adult *C. hemapterus* flies find a nest for parasitism, they lose their wings and feed on incubating adults and growing nestlings [35,36].

The hoopoes used artificial nest-boxes for breeding in the study area, and the experiment consisted of installing new nest-boxes filled with old nest-material remains collected from nest boxes where hoopoes bred the previous year. Nest materials from previous years were autoclaved in order to eliminate the microorganisms in comparison to control, nonautoclaved nest materials. We expect that the hoopoes breeding in experimental nest boxes, where symbiotic bacteria of added nest-materials were eliminated, would experience a lower probability of ectoparasitism than those nesting in control nest-boxes. Moreover, since ectoparasitism has negative effects on nestling growth, we predicted that the nestlings in control nests will experience lower fledging success than those growing in experimental nests. This prediction is based on the assumption that experimental nest boxes with autoclaved nest-material should have bacteria at a lower density than control nest-boxes at the time of hoopoes’ reproduction.

## 2. Materials and Methods

### 2.1. Study Area and Species

The fieldwork was carried out during the 2017 and 2018 breeding seasons (March–July) on hoopoe nests in a wild population located in the Hoya de Guadix (Granada, Southern Spain, 37°18′ N, 38°11′ W), a plateau at 1000 m above sea level and semiarid climate. In this area, around 300 cork-made nest-boxes are available for hoopoes; most of them attached to tree trunks and walls but also hidden in piled stones. The dimensions of nest-boxes are 35 × 18 × 21 cm (internal height × width × depth), 24 cm (bottom-to-hole height), and 5.5 cm (entrance diameter). Information about the study area is further described elsewhere [27,37].

The hoopoe is a Upupiformes migratory species, distributed throughout Europe, Asia, and Africa [38,39]. Hoopoes are hole-nesters that frequently use natural cavities, trees, walls, or artificial nest-boxes for reproduction. They do not build nests but prefer cavities with remains of soft material from previous reproduction events of conspecifics or heterospecifics, where they create a small hole and lay the clutch [40]. In our study area, the reproductive season starts in late February, and many females lay a second clutch before August.

We assessed ectoparasitism by *C. hemapterus,* a common generalist hematophagous fly of about 2 mm in length that is easily detected on the nestling body. *C. hemapterus* is a frequent ectoparasite in hoopoe nests in our study area [32]. Following previously published work [29], the intensity of parasitism was inferred from the abundant traces of blood remains and feces on nestlings’ skin (i.e., belly and wing; see below).

### 2.2. Experimental Design and Fieldwork

The experiment was performed in new nest-boxes installed before the start of hoopoe reproduction in both breeding seasons (2017 and 2018). Before reproduction started, at the beginning of February, we visited nest-boxes where hoopoes successfully bred the year before and collected old nest-materials that were stored in punched plastic bags (47 nest-boxes in 2017 and 70 in 2018) and maintained at room temperature. Nest-materials from different nest-boxes were pooled, mixed, and divided into two halves: one of them was autoclaved (to be used in experimental nest boxes), and the other was not (to be used in control nest boxes). *C. hemapterus* pupae overwinter inside bird nests, and autoclaving the nest material should kill them. Therefore, after treatment, experimental and control material could differ not only in bacterial density but also in the probability of *C. hemapterus* flies emerging within the control nest boxes, which might affect experimental outcomes. To evaluate this potential bias, we removed, in 2018, all pupae from the material of both experimental treatments before autoclaving. We did so by sifting nest-material with opening meshes of 2, 1, and 0.5 cm diameters [41]. Nest-boxes used in this experiment were new and, before installation, each of them was filled with 500 cm^3^ of experimental nest-material (autoclaved or nonautoclaved according to the assigned experimental treatment) mixed homogenously with 500 cm^3^ of commercial sawdust (Allspan^®^ Animal bedding, Moult, France, wood shavings). As new nest-boxes replaced old ones, experimental treatments were sequentially alternated, and the appropriate nest-material added. This procedure was performed wearing new latex gloves for each nest-box to avoid cross-contamination between experimental and control nests materials. A total of 86 and 69 new nest-boxes were arranged in the study area in 2017 and 2018, respectively.

Nest-boxes were inspected every four days, from early March to the end of June, which allowed us to estimate the start of laying when a nest-box was detected with hoopoe eggs. Nest-boxes were visited again 17 days after the start of laying and daily afterward to detect the hatching date, which is expected to occur 18 days after the start of incubation [40]. Four days after the first egg hatched, we collected nest-material in contact with chicks for bacterial analyses. Samples were stored in previously sterilized 1.5 mL microfuge tubes and kept in a portable fridge at 4 °C until being processed in the laboratory within the following 12 h. Nest-boxes were again visited when the older nestlings were eight and nineteen days old. During these two last visits, we measured the body mass of all nestlings with a hanging scale (Pesola 0–50 and 0–100 g, depending on the nestling age, accuracy 1 g), and estimated the intensity of ectoparasitism by *C. hemapterus* as the number of spots due to blood remains on the skin of the belly and left underwing of nestlings. We performed all manipulations with new latex gloves cleaned with 70% ethanol to prevent cross-nest contamination.

### 2.3. Laboratory Procedures

Nest bacterial density was estimated by means of traditional culture methods. Briefly, under sterile conditions, we included 1 cm^3^ of nest-material in a sterile Falcon tube with 1 mL of sterile sodium phosphate buffer (PBS, 0.2 Molar; pH = 7.2). The Falcon tubes were vigorously vortexed to homogenize samples. Serial dilutions up to 10^−4^ were cultivated by spreading 5 microliters of each dilution in plates with Tryptone soya agar (TSA), a broad medium used to grow mesophilic bacteria. The plates were incubated aerobically at 37 °C for 24 h before colony counting. Bacterial counts were performed in plates of the serial dilution with around 30–300 colonies. Estimates of bacterial loads were standardized to the number of colony-forming units (CFU) per mL (number of colonies × 10 ^dilution factor^)/0.005 mL spread).

### 2.4. Statistical Analyses

We were interested in detecting expected influences of autoclaved nest-material from previous reproduction on hoopoe nestling parasitism and, thus, we only considered the first breeding attempt detected in each of the experimental nest-boxes.

Nest bacterial density at the early nestling stage (4 days after hatching) approached a normal distribution after log transformation (Kolmogorov–Smirnov test for continuous variables, *p* > 0.05). The intensity of the ectoparasitism of nestlings eight days after the first egg hatched followed approximately a Gaussian distribution after square root transformation (Kolmogorov–Smirnov test for continuous variables, *p* > 0.05). Fledging success, estimated as the number of nestlings alive on the second visit divided by the brood size at the first visit, was not transformed before analyses and was used in percentage. Residuals of all statistical models followed normal distributions.

The effect of experimental treatment on nest bacterial density was explored in a general linear model (GLM) that also included the study year as a fixed discrete factor and laying date and number of nestlings as covariables. The experimental effect on ectoparasitism intensity was explored in a similar GLM that also included nest-material bacterial loads as an additional independent covariable because the hypothesis tested posits that parasitism should be mediated by the nest’s bacterial environment. Finally, the experimental effect on fledging success was explored in a similar GLM that also included the intensity of parasitism as an additional independent covariable. For all these GLMs, we estimated the variance explained by the whole model (R^2^) and first-order interactions between experimental treatments and all other independent factors. These interactions were explored in separate models that also included main effects. In no case did brood size explain a significant proportion of variance of dependent variables, either alone or in interaction with experimental treatment (results not shown). Moreover, considering brood size as an additional independent factor did not influence statistical inferences qualitatively (i.e., statistical significance; results not showed). Thus, we herein present results from models that did not include brood size as an independent factor.

All these analyses were performed in the R environment (R 3.4.3, https//www.r-project.org/) with library “glm” [42].

## 3. Results

During reproduction, at the early nestling stage, hoopoe nest-boxes with nest-material that was autoclaved before reproduction started tended to harbor lower bacterial densities than hoopoe nest-boxes that were filled with control nest material. That was the case after controlling for the statistically significant effects of the laying date and the study year (Table 1). Interestingly, the detected trend (although nonsignificant) was consistent between study years (see interaction in Table 1).

With respect to ectoparasitism by *C. hemapterus* flies, hoopoe nestlings developing in nest-boxes with autoclaved material presented lower ectoparasite loads than those in nest-boxes with control material (Table 1). This experimental effect depended on the study year and nest bacterial loads (Table 1). The effect was stronger in 2018 (Figure 1), when the expected positive association between ectoparasitism and nest bacterial load appeared exclusively in experimental nests (Figure 1). These patterns were detected after controlling for the positive association between laying date and parasitism intensity (Figure 1). The expected effects in parasitism intensity were, therefore, more clearly detected in the year when *C. hemapterus* pupae were removed from nest-materials.

Finally, fledging success of experimental and control nests did not differ significantly (Table 1), but the bacterial density of hoopoe nests, eight days after hatching, was positively associated with fledgling success, even after controlling for the nonsignificant effects of experimental treatment, study year, laying date, and intensity of parasitism (Table 1). This positive association was also detected after excluding from the models the nonsignificant variables.

## 4. Discussion

Our main results show that autoclaving nest-material reduced the intensity of ectoparasitism suffered by nestling hoopoes. Moreover, in the case of nest-boxes with autoclaved nest-material, but not in the control nest-boxes, ectoparasitism intensity was positively related to bacterial density. In addition, the bacterial density of hoopoe nest-material was positively related to fledging success. Finally, we found that nest-boxes with experimental autoclaved nest-material tended to have lower bacterial density than those with control nest-material, although not significantly. All these results, considered together, suggest a role for the bacterial environment due to nest-material influencing ectoparasitism and fledging success in hoopoes. Below, we discuss this inference, alternative hypotheses, and possible mechanisms that could fit our experimental results.

Ectoparasites might use different sensory channels, such as visual, auditory, or olfactory, to detect nests of their avian hosts [43,44]. However, *C. hemapterus*, the main blood-sucking ectoparasite of hoopoes in our study area, mainly parasitizes hole-nesting birds and, thus, the most likely theory is that ectoparasites detect host nests follow chemical cues. Although we have not measured chemical volatiles associated with experimental hoopoe nests, our results, including the detected experimental effect on the intensity of ectoparasitism, and the association between bacterial density of nest-material remains and intensity of ecto-parasitism suggest that chemical cues hypothetically used by ectoparasites should be mediated by bacterial metabolism.

The detected effect of autoclaving nest-material remains on ectoparasitism intensity might be directly mediated by its effect on bacterial densities of nest-material, or indirectly by the effect of experimental nest-material on the bacterial community of the uropygial gland of hoopoe females. The uropygial gland of incubating or brooding hoopoe females is full of volatile-producing bacteria [26] that *C. hemapterus* flies might use to detect active nests. Our experiment might have affected the bacterial community of the uropygial gland and hence the amount and diversity of volatiles in the nest-box and surroundings. However, previously published experimental evidence strongly suggests that the presence of nest-material remains from previous breeding seasons influences the bacterial community of the uropygial secretion of females [27]. Therefore, given the strong association between the bacterial community and the volatile profile of uropygial secretions [26], it is likely that the detected effect on the intensity of ectoparasitism was mediated by changes produced in the bacterial community of the uropygial gland. The effect of the nest’s bacterial environment on the bacterial community of the uropygial gland surroundings [45] or in the volatile profiles of the secretion [46] has also been detected in dark-eyed juncos (*Junco hyemalis*) and great tits (*Parus major*), respectively. Thus, it is possible that the detected influence of the nest-material bacterial community on ectoparasitism intensity in hoopoes could be extrapolated to other bird species.

Our experimental approach assumed that autoclaving nest-material remains before reproduction started should affect the nest bacterial community at the nestling stage. However, although bacterial density in nest-material of nest-boxes under the control treatment tended to be higher than that of nest-boxes under the experimental treatment, the differences only approached statistical significance. It is worth to mention here that using a similar experimental approach, Díaz-Lora and coauthors [27] found that bacterial loads on hoopoe eggshells in nest-boxes with nest bacterial remains mixed with sawdust was significantly higher than that of eggshells in nest-boxes with only sawdust. Thus, it is possible that the expected experimental effect on nest bacterial density was more easily detected at the egg stage. Contrary to most avian species [47], hoopoe parents do not remove fresh nestling feces from nests, which could produce an increase in bacterial density after hatching, as has been shown for starlings [21]. Bacterial samples were collected 4 days after the egg hatched and, thus, the effects of autoclaving nest-material could have been masked or diluted by the nestlings’ activity. Information on bacterial communities of nest-material, of the uropygial secretion of females, and on the associated volatile profiles are, however, necessary to explore the possibility that it explains the detected experimental effect on ectoparasitism intensity.

In a similar experimental approach, Podofillini and coworkers [41] used new nest boxes where they included sand collected from the nest surroundings that was or was not mixed with old nest-material from previous reproduction of lesser kestrel (*Falco naumanni*). Consistent with our results, they found out that nestlings living in nests with materials from previous breeding seasons suffered from *C. hemapterus* more than those living in nest-boxes filled only with sand. However, these results do not necessarily have to be the consequence of bacteria in the experimental nest-material, but rather of the emergence of adult ectoparasites from overwintering pupae in the used nest material. Trying to differentiate between these two possibilities, we screened nest-material collected during 2018 and eliminated all *C. hemapterus* pupae. In 2017, when we did not eliminate *C. hemapterus* from nest materials, levels of parasitism were similar between experimental and control groups. The effects of volatiles from nest-material, but also those from *C. hemapterus* emergence from nest-material that was not autoclaved nor screened for parasites pupae, should influence parasitism positively; differences between control and experimental nests should therefore be higher in 2017. However, contrary to this possibility, the expected experimental effect was clearer in the year in which *C. hemapterus* pupae were removed from nest-materials (i.e., 2018). Thus, it is unlikely that overwintering *C. hemapterus* pupae in nest-material remains contributed to explaining our results. Consequently, the higher intensity of ectoparasitism of nestlings in control nests in 2018 is the consequence of the experimental treatment.

Interestingly, and consistent with the role of bacteria explaining the ectoparasitism of nestlings, nest-material bacterial loads and ectoparasitism intensity was positively related. Furthermore, although the strength of this association was similar in 2017, the association between nest-material bacterial load and parasitism was absent for control but not for experimental nests in 2018. We do not have a plausible explanation for this result. We only may speculate with the possibility that *C. hemapterus* take their cue from volatiles of particular bacterial strains that hoopoe females acquire when breeding in nests-boxes with material remains from previous reproductions [27]. This is a possibility to be explored in future works. An alternative possibility is that parasite activity influences the bacterial density of hoopoe nests. We know that *C. hemapterus* parasitize incubating birds and that its activity influences eggshell bacterial loads positively [29]. Consequently, higher bacterial density would be expected in the nests where *C. hemapterus* feeding on nestlings are more abundant. Another alternative hypothesis explaining our results is related to the possibility that *C. hemapterus* follows chemical cues from flies already present in hoopoe nests. However, for one of the study years, we removed *C. hemapterus* pupae from all collected nest-material and nest-boxes were newly installed; hence, none of the parasitizing flies in experimental or control nests came from nest material that study year. Thus, because all parasitizing flies in our nest boxes should have come from other nests, chemicals from previous *C. hemapterus* activity would hardly explain the detected experimental effects in ectoparasitim intensity. Thus, experimental manipulation of *C. hemapterus* flies in hoopoe nests is necessary to further conclude that bacteria are the cause of the detected association with parasitism intensity.

Contrary to our expectation, fledging success and bacterial loads of nest-material was positively associated. Nest bacterial loads are usually negatively related to breeding success in birds [48,49,50] but not in hoopoes [27], which might be related to the special lifestyle of this species. Hoopoes do not remove nestling feces and harbor symbiotic bacteria in their uropygial gland at a high density [51]. Some of the symbionts produce antibiotic substances [22,23,24,25,51] that might protect growing nestlings and, thus, we speculate with the possibility that these antibiotic-producing bacteria were more abundant in nests with higher bacterial density and explained the positive detected association. Experimental modification of the uropygial gland bacterial community (e.g., by injecting antibiotics [26]) is, however, necessary to explore this possibility. 

## 5. Conclusions

In summary, our experimental results linking the nest bacterial communities and ectoparasitism suggest that *C. hemapterus* flies follow chemical cues to find active nests of their hoopoe hosts and that these chemical volatiles are produced by bacteria growing in nest-material or in the uropygial gland of females. Characterization of the volatile profiles of hoopoe nests and secretions, as well as those of the associated bacterial communities of experimental and control nests, are necessary to further explore mechanisms explaining the detected lower parasitism intensity of nestling developing in nests with autoclaved nest-materials.

## Figures and Tables

**Figure 1 biology-09-00306-f001:**
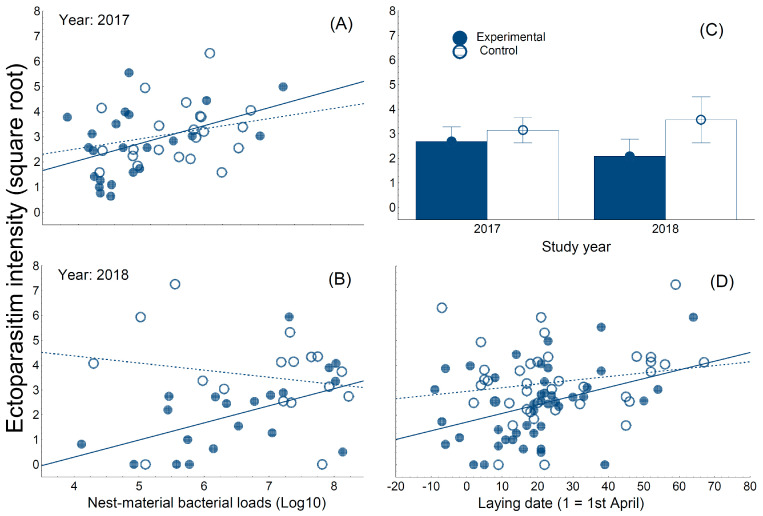
Associations between the ectoparasitism intensity of hoopoe nestlings and bacterial loads of nest-materials that were (filled dots) or were not (open dots) autoclaved before reproduction started in the 2017 (**A**) and 2018 (**B**) study years. We also show weighted means (±95% CI) of ectoparasitism intensity suffered by hoopoe nestlings developing in nest-boxes with nest-material that were or were not autoclaved in the 2017 and 2018 breeding seasons (**C**). Finally, (**D**) we also show the association between ectoparasitism intensity and laying date of hoopoe nests with autoclaved or not autoclaved nest-material. Solid and dashed lines are regression lines for experimental and control nest-boxes.

**Table 1 biology-09-00306-t001:** Results from general linear models exploring the effect of autoclaving nest-material (Treatment) of nest-boxes that hoopoes bred on: log10-transformed bacterial loads of experimental nest-boxes during hoopoes’ reproduction, the intensity of parasitism (i.e., square root transformed number of bites), fledging success (percentage).

Predictors	Mean1 (SE)	Mean2 (SE)	β (SE)	Main Effects			Interaction with Experimental Treatment		
				F	df	P	F	df	P
**Bacterial loads of nest-material (R^2^ = 0.49, F = 29.86, df = 3, 94; *p* < 0.001)**									
Treatment	**5.57 (0.18)**	**5.96 (0.17)**		**3.31**	**1, 94**	**0.072**			
Year	**5.05 (0.09)**	**6.68 (0.18)**		**63.35**	**1, 94**	**0.001**	0.23	1, 92	0.630
Laying Date			**0.01 (0.01)**	**4.90**	**1, 94**	**0.029**	0.94	1, 92	0.760
**Intensity of Ecto-parasitism (R^2^ = 0.20, F = 4.90, df = 4, 79, *p* = 0.001)**									
Treatment	**2.39 (0.22)**	**3.33 (0.23)**		**4.77**	**1, 79**	**0.031**			
Year	**2.91 (0.18)**	**2.75 (0.29)**		**3.30**	**1, 79**	**0.073**	**8.31**	**1, 76**	**0.005**
Laying Date			**0.02 (0.01)**	**6.05**	**1, 79**	**0.016**	0.78	1, 76	0.378
Bact Loads			0.24 (0.17)	2.12	1, 79	0.149	**5.21**	**1, 76**	**0.025**
**Fledging Success (R^2^ = 0.06, F = 2.04, df = 5, 78, *p* = 0.083)**									
Treatment	70.25 (4.45)	59.95 (4.90)		0.55	1, 78	0.462			
Year	59.39 (4.49)	72.92 (4.74)		0.00	1, 78	0.962	0.96	1, 74	0.331
Laying Date			−0.20 (0.15)	1.82	1, 78	0.181	1.66	1, 74	0.202
Bact Load			**6.01 (2.53)**	**5.63**	**1, 78**	**0.020**	0.03	1, 74	0.855
Int Parasitism			−0.89 (1.68)	0.29	1, 78	0.596	0.46	1, 74	0.501

For each model we show variance explained (R^2^), F statistics, degrees of freedom (df) and *p*-values. We also show least square means (SE) for experimental (Mean 1) and control (Mean 2) nests (first line of each model tested), and for the years 2017 (Mean 1) and 2018 (Mean 2) (second line of each model tested). For continuous predictors (laying date, nest-material bacterial load (Bact Load), and intensity of ectoparasitism (Int parasitism), we show beta (SE) values. The first-order interactions between experimental treatment and the other independent factors were tested in separate models that also included main effects, while the main effects were explored in models that did not include interactions. Statistical effects with associated alpha-values lower than 0.1 are highlighted in bold fonts.

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
