# Peer review of "Autoclaving Nest-Material Remains Influences the Probability of Ectoparasitism of Nestling Hoopoes (Upupa epops)"

_biology, 2020, doi:10.3390/biology9100306_

Round 1

Reviewer 1 Report

I think the main result of the investigation are the relationships between autoclaving nest-material and intensity of ecto-parasitism of nestlings. It was also interesting that bacterial density of hoopoe nest-material was positively related to fledging success.

I think the manuscript should be shorten (maybe the short communication can be prepared), especially with the view of introduction (lines 51-70 seems to be a bit out of topic) and discussion (lines 123-127 are not methods, but discussion, information about begging calls is also out of topic). Authors talk a lot about „bacterial community“, but only the bacterial density was investigated in the manuscript. The volatile profile of the experimental nest-boxes was also not investigated, so the discussion can be shortened.

I would also suggest not use supplementary Table S1.

There are some repetitions in the manuscript. For example, information in line 87 and line 119-121 is the same; line 111 and lines 140-142; lines 100-101 and 145-146.

I would suggest instead of „Carnus“use entomological traditional „C. hemapterus“.

Line 144 – what does mean year 2016?

Line 150 - „...before autoclaving, we removed all pupae from material of both experimental groups in 2018“. What did You do in 2017? Autoclaved with the pupas?

Line 151 – „...with opening meshes of 2, 1 and 0.5 cm diameters...“ What was the use of three different diameters?

Line 151 – 153 – It should be explained more clearly does the filling the nest-box with nest-material / sawdust was applied for all nest boxes (experimental and control) or not.

Line 161-163 – „Nest-boxes were sampled the fourth and nestlings were sampled the eighth and the nineteenth day after the first egg hatched. During these two visits...“ I have found not two, but three visits. What does mean „During the first visit with nestlings, we also collected nest-materials in contact with chicks (lines 164-165)“ the fourth or the eighth day after hatching?

Line 249-251. You wrote: „Finally, we found that nest-boxes with experimental autoclaved nest-material tended to have lower bacterial density than those with control nest-material; although not significantly.“

But Lines 208-209 You wrote opposite: „Hoopoe nests with autoclaved old nest-materials tended to harbour lower bacterial densities than control nests during reproduction, at the early nestling stage.“

These to sentences seems to oppose each other.

In common the manuscript is interesting, only I would prefer it to be shorten and presented as short communication.

Author Response

Response to Reviewer 1 Comments

I think the main result of the investigation are the relationships between autoclaving nest-material and intensity of ecto-parasitism of nestlings. It was also interesting that bacterial density of hoopoe nest-material was positively related to fledging success.

Point 1: I think the manuscript should be shorten (maybe the short communication can be prepared), especially with the view of introduction (lines 51-70 seems to be a bit out of topic) and discussion (lines 123-127 are not methods, but discussion, information about begging calls is also out of topic). Authors talk a lot about „bacterial community “, but only the bacterial density was investigated in the manuscript. The volatile profile of the experimental nest-boxes was also not investigated, so the discussion can be shortened.

Response 1:

  • Following referee suggestion, we have shortened the referred paragraph in the Introduction (new lines 51-61), and moved from Material and Methods to the Introduction section the lines describing uropygial secretion of female and nestling hoopoes. New lines 79-84.
  • It is true that the only descriptive measure of the bacterial community of hoopoe nest we measured was bacterial density. For the new version, we mostly refer to this measure rather than to bacterial community in general.
  • We have also reduced the text in the discussion section dealing with begging calls and volatile profiles which, as the reviewer mention, we did not measure, but assume that our experiment will affect bacterial metabolism (new lines 247-249).

Point 2: I would also suggest not use supplementary Table S1.

Response 2: We have removed Table S1 from the new version and only mentioned in the material and methods that “Results do not vary qualitatively when including information of brood size in the analyses (results not shown).”. New lines 189-192.

Point 3: There are some repetitions in the manuscript. For example, information in line 87 and line 119-121 is the same; line 111 and lines 140-142; lines 100-101 and 145-146.

Response 3: Thanks, for the new version, we have removed lines 87 and 145-146. Lines 140-142 are however necessary since we there explain that we used new next boxes and that the experiment was performed during the two breeding season.

Point 4: I would suggest instead of „Carnus“use entomological traditional „C. hemapterus“.

Response 4: OK, we have replaced Carnus by C.hemapterus throughout the manuscript.

Point 5: Line 144 – what does mean year 2016?

Response 5: We refer to the year when hoopoes reproduced on the collected material. Trying to avoid possible misleading, in the new version we refer to the year when we collected nest material (2017-2018). Lines 128-131:

Point 6: Line 150 - „...before autoclaving, we removed all pupae from material of both experimental groups in 2018“. What did You do in 2017? Autoclaved with the pupas?

Response 6: Yes, we autoclaved the nest material with the pupas in 2017.

Point 7: Line 151 – „...with opening meshes of 2, 1 and 0.5 cm diameters... “What was the use of three different diameters?

Response 7: It is the usual protocol because it removes grains of different size allowing to separate material of similar size to that of pupae where they are more easily isolated from grains of larger and smaller size.

Point 8: Line 151 – 153 – It should be explained more clearly does the filling the nest-box with nest-material / sawdust was applied for all nest boxes (experimental and control) or not.

Response 8: Yes. We have re-written the sentence to be more clear.

New lines 139-142: “Nest -boxes used in this experiment were new and, before installation, each of them was filled with 500 cm3 of experimental nest-material (autoclaved or non-autoclaved according to the assigned experimental treatment) mixed homogenously with a 500 cm3 of commercial sawdust (Allspan® Animal bedding, wood shavings).”.

Point 9: Line 161-163 – „Nest-boxes were sampled the fourth and nestlings were sampled the eighth and the nineteenth day after the first egg hatched. During these two visits...“ I have found not two, but three visits. What does mean „During the first visit with nestlings, we also collected nest-materials in contact with chicks (lines 164-165) “the fourth or the eighth day after hatching?

Response 9: Trying to clarify the text, we have rewritten these lines.

New lines 150-157:Four days after the first egg hatched, we collected nest-material in contact with chicks for bacterial analyses. Samples were stored in previously sterilized 1.5 ml microfuge tubes and kept in a portable fridge at 4ºC until being processed in the laboratory within the following 12 hours. Nest-boxes were again visited when the older nestling was eight and nineteen days old. During these two last visits, we measured body mass of all nestlings with a hanging scale (Pesola 0-50 and 0-100 g, depending of the nestling age, accuracy 1 g), and estimated the intensity of ecto-parasitism by C. hemapterus as the number of spots due to blood remains on the skin of the belly and left under-wing of nestlings.”.

Point 10: Line 249-251. You wrote: „Finally, we found that nest-boxes with experimental autoclaved nest-material tended to have lower bacterial density than those with control nest-material; although not significantly.

But Lines 208-209 You wrote opposite: „Hoopoe nests with autoclaved old nest-materials tended to harbour lower bacterial densities than control nests during reproduction, at the early nestling stage.

These two sentences seem to oppose each other.

Response 10:  In both cases, we wanted to express that nest boxes with material that was previously autoclaved presented lower bacterial densities than nest boxes with control non-autoclaved material. We have modified wordiness.

New text lines 197-201: “During reproduction, at the early nestling stage, hoopoe nest boxes with nest-material that was autoclaved before reproduction started tended to harbour lower bacterial densities than hoopoe nest-boxes that were filled with control nest-material.”.

Point 11: In common the manuscript is interesting, only I would prefer it to be shorten and presented as short communication.

Response 11: We have tried to shorten the manuscript by following the referee's guidance.

Reviewer 2 Report

The authors used an experimental approach and autoclaved nesting material to remove bacteria, and then tested the association between bacteria load and number of Carnus hemapterus  ectoparasites per nest box of Hoopoes. They also measured the association between bacteria load and fledging success. In 2017, the nest material was autoclaved but the previous clutches of Carnus pupae were not removed. In 2017, bacteria load (+ residual Carnus pupae) was associated with number of ectoparasites but there was no significant effect of treatment. In 2018, the nest material was autoclaved and Carnus pupae were removed. In 2018, bacteria load predicted number of ectoparasites in the treatment group but not the control group. This suggests an interaction effect of initial number of Carnus pupae with bacteria load on the pattern. The authors conclude that “Carnus flies follow chemical cues to find active nests of their hoopoe nests, and that these chemical volatiles are produced by bacteria growing in nest material” – which overstates their findings, as it is possible that Carnus use cues produced by other Carnus. The authors do not mention the possibility of chemical signaling by Carnus. More attention needs to be given to potential chemical signaling between Carnus ectoparasites who may be competing for particular hosts or who may have an optimal density per nest box. You found a signal of effect from manipulating the number of initial Carnus pupae so the conclusion cannot be entirely based on the nest material bacteria.

Is it possible to count the puparia cases to estimate the residual Carnus infection level? You could calculate residual and new Carnus infections by counting the number of empty puparia cases versus active pupae.

Given the potential interaction effect between residual Carnus and bacteria load, the conclusion could be toned down a bit:

Lines 341-342: “…Carnus flies follow chemical cues to find active nests of their hoopoe nests, and that these chemical volatiles may be produced by bacteria growing in nest material”. It is equally plausible that the chemical cues are from the Carnus ectoparasites themselves, who may be signaling nest occupancy etc. From a female Carnus perspective, she may want to avoid nests with particular Carnus age cohorts and so on.

Do parasitized nestlings have different bacteria profiles in their fecal sacs compared with non-parasitized nestlings?

The English is awkward and sometimes incorrect, and so this should be proof-read by a native speaker.

This is a very nice study!

Author Response

Point 1: The authors used an experimental approach and autoclaved nesting material to remove bacteria, and then tested the association between bacteria load and number of Carnus hemapterus ectoparasites per nest box of Hoopoes. They also measured the association between bacteria load and fledging success. In 2017, the nest material was autoclaved but the previous clutches of Carnus pupae were not removed. In 2017, bacteria load (+ residual Carnus pupae) was associated with number of ectoparasites but there was no significant effect of treatment. In 2018, the nest material was autoclaved and Carnus pupae were removed. In 2018, bacteria load predicted number of ectoparasites in the treatment group but not the control group. This suggests an interaction effect of initial number of Carnus pupae with bacteria load on the pattern. The authors conclude that “Carnus flies follow chemical cues to find active nests of their hoopoe nests, and that these chemical volatiles are produced by bacteria growing in nest material” – which overstates their findings, as it is possible that Carnus use cues produced by other Carnus. The authors do not mention the possibility of chemical signaling by Carnus. More attention needs to be given to potential chemical signaling between Carnus ectoparasites who may be competing for particular hosts or who may have an optimal density per nest box. You found a signal of effect from manipulating the number of initial Carnus pupae so the conclusion cannot be entirely based on the nest material bacteria.

Response 1: We really appreciate the proposed alternative hypothesis. See below our responses to particular comments.

Point 2: Is it possible to count the puparia cases to estimate the residual Carnus infection level? You could calculate residual and new Carnus infections by counting the number of empty puparia cases versus active pupae.

Response 2: Unfortunately, we do not have this kind of information.

Point 3: Given the potential interaction effect between residual Carnus and bacteria load, the conclusion could be toned down a bit:

Response 3: Although it is possible that Carnus follow chemical cues from flies already present in hoopoe nests, in our modest opinion, it would hardly explain our results. First, because, for one of the study years, we removed Carnus pupae from all collected nest material. And, thus, since nest boxes were newly installed, none of the parasitizing Carnus in experimental or control nests came from nest material that study year (2018). Moreover, this was the year when the association between bacterial density and intensity of ecto-parasitism was primary detected. Thus, because all parasitizing in our nest boxes should have come from other nests, Carnus chemicals from previous Carnus activity would hardly explain the detected experimental effects in ecto-parasitim intensity. In the new version we have discussed the alternative explanation proposed by the reviewer. New lines 315-322.

Point 4: Lines 341-342: “…Carnus flies follow chemical cues to find active nests of their hoopoe nests, and that these chemical volatiles may be produced by bacteria growing in nest
material”. It is equally plausible that the chemical cues are from the Carnus ectoparasites themselves, who may be signalling nest occupancy etc. From a female Carnus perspective, she may want to avoid nests with particular Carnus age cohorts and so on.

Response 4: The association between bacterial density and intensity of Carnus parasitism were mainly detected the study years, when Carnus pupae were remove from collected nest- material before using it to fill installed new next-boxes. Thus, even if chemicals from Carnus activity attracts or repel winged flies it would hardly explain the detected experimental effects.

Point 5: Do parasitized nestlings have different bacteria profiles in their fecal sacs compared with non-parasitized nestlings?

Response 5: hoopoes do not used fecal sacs covering faeces, and parents do not remove feces from nest holes. We do not know whether the experiment affect intestinal microbiota, but as shown by a previous work (Diaz Lora et al. 2019 J Avian Biol), it affected microbiota of the uropygial secretion and on the eggshells.

Point 6: The English is awkward and sometimes incorrect, and so this should be proof-read by a native speaker.

Response 6: we have carefully correct all typos and done our best.

Point 7: This is a very nice study!

Response 7: Thank you very much.

Reviewer 3 Report

The manuscript by Mazorra Alonso and colleagues describes the influences of bacteria and ectoparasitism on nestling hoopoes.

The study has been designed and described well and the data are sound and based on extensive statistical analysis.

Major comments:

The abstract is structured and clearly presented and describes the scope of the paper well.

The introduction presents the scope of the research very well and puts in clearly in the literature context.

Comments; English grammar should be reviewed, in particular singular and plural verb use. I will not mention them here, because there is quite a number of them; please review the text critically. Line 46, recently: this refers to reference 6, but certainly not to 5 (2011). The information therefore is not that recent and got a follow-up in 2019. Consider rephrasing the sentence. Line 48: bacteria should be bacterial. Line 50: kindship should kinship. Line 61, volatiles should be singular (volatile compounds). Line 67: evidences should be evidence. Line 78, the reference indicated should be numbered (36). Line 92: volátiles, change to volatiles. Line 96: These findings: which ones? I would change this to The expected findings. Lines 100/101. I would rephrase the sentence and mention the autoclaved nests. Something like “Nest materials […] were autoclaved in order to eliminate the microorganisms [……] in comparison to control, non-autoclaved nests”.

Materials and methods.

Rewrite line 111 in the same way as 141 (2017-2018 vs breeding seasons 2017 and 2018). It is more clear that way that you are talking about different breeding seasons.

Line 114; truck should be trunk.

Reverse the order of words in line 119: Hoopoes are hole-nesters that frequently use natural cavities, trees or walls or artificial next-boxes for reproduction.

The information presented in 2.1 is kind of mixture of introductory information and area/species information. I would transfer parts of lines 123-127 and of lines 131-138 to the introduction, because it is more introduction than materials and methods.

Line 147: are you sure about the killing of the Carnus pupae by autoclaving? Is there a reference? And, in addition, did you check if the pupae were removed by the procedure described in lines 150 and 151?

Line 171: by means of.

Results section.

In general; to my opinion this part is a bit weak and not balanced with the rest of the manuscript (other sections have far more text). I would suggest to say more about the data, also in relation to the next comment.

On the order of data presentation: why not explain first figure 1 and then table 1? Or, put the table first in the manuscript. Readers would be inclined to look at the figure first. On this figure 1; the data presentation would benefit from minor changes: for consistency reasons, draw the uninterrupted line representing the trend for the closed circles and the interrupted line for the open circles. It is not logical this way. Second, the relevance of the connection in 1C is not clear to me. Why not show these data in bar graph form? Or please explain the line in the figure. The figure now suggests a relationship between different breeding seasons. Figure 1D: where in the figure can we see the ‘1’? I get the point, but maybe it is more logical to say that 0 is the start of the breeding season starting at the beginning of April.

About table 1: I would need more explanation. It is not clear to me for instance if the Mean numbers represent experimental versus controls or if they represent 2017 versus 2018 situations. I am kind of confused by the fact that ‘year’ is also mentioned as a variable on the left side in the table. Is there a way to make this information more clear? Minor comment: using comma and dots in the data table does not look very consistent.

Minor comment: table 2 is referred to as table S1. And, it contains more or less the same information with additional information. Would it not be logical to show only new information or merge both tables?

Discussion.

The discussion would benefit from opening with a general remark on the work which introduces the topics to be discussed. The first sentence, line 246 is a bit awkward. Use words like ‘the treatment affected the intensity (or the number) of’. I understand that the presence of Carnus may influence results and that this issue was sorted out in the current study. To make lines 305 a bit more clear (please, have a look at the English here again), make a clear discrimination between what was done in which year. Since the studies described were done in 2017 and 2018, I assume that you are talking about non-published work in 2017 which is contrasting the results from the current study in which Carnus pupae were removed. This might be stated more clearly. If the above is not the case, then write in the materials and methods that experimentation was done different in 2017 compared with 2018.

Lines 338 and 339: Experimental modification is necessary. How do you suggest to do this? And also, can you make suggestions on the methodology of profiling mentioned in lines 343 and further?

Minor comments:

Quite some typographical mistakes in the different sections.

Author Response

Response to Reviewer 3 Comments

Comments and Suggestions for Authors

The manuscript by Mazorra Alonso and colleagues describes the influences of bacteria and ectoparasitism on nestling hoopoes.

The study has been designed and described well and the data are sound and based on extensive statistical analysis.

Major comments:

The abstract is structured and clearly presented and describes the scope of the paper well.

The introduction presents the scope of the research very well and puts in clearly in the literature context.

Point 1: Comments; English grammar should be reviewed, in particular singular and plural verb use. I will not mention them here, because there is quite a number of them; please review the text critically. Line 46, recently: this refers to reference 6, but certainly not to 5 (2011). The information therefore is not that recent and got a follow-up in 2019. Consider rephrasing the sentence. Line 48: bacteria should be bacterial. Line 50: kindship should kinship. Line 61, volatiles should be singular (volatile compounds). Line 67: evidences should be evidence. Line 78, the reference indicated should be numbered (36). Line 92: volátiles, change to volatiles. Line 96: These findings: which ones? I would change this to The expected findings. Lines 100/101. I would rephrase the sentence and mention the autoclaved nests. Something like “Nest materials […] were autoclaved in order to eliminate the microorganisms [……] in comparison to control, non-autoclaved nests”.

 Response 1: Thank you very much for all these typos corrections. We have incorporated all of them in the new version. Moreover, we have carefully and critically read the whole manuscripts and correct all detected typos.

 Point 2: Materials and methods.

Rewrite line 111 in the same way as 141 (2017-2018 vs breeding seasons 2017 and 2018). It is more clear that way that you are talking about different breeding seasons.

 Response 2: Rewritten. New lines 108 and 127-128.

Point 3: Line 114; truck should be trunk.

Response 3: Replaced

Point 4: Reverse the order of words in line 119: Hoopoes are hole-nesters that frequently use natural cavities, trees or walls or artificial next-boxes for reproduction.

Response 4: Done. New lines 116-117.

Point 5: The information presented in 2.1 is kind of mixture of introductory information and area/species information. I would transfer parts of lines 123-127 and of lines 131-138 to the introduction, because it is more introduction than materials and methods.

Response 5: We transferred these lines to the Introduction section. New lines78-94.

Point 6: Line 147: are you sure about the killing of the Carnus pupae by autoclaving? Is there a reference? And, in addition, did you check if the pupae were removed by the procedure described in lines 150 and 151?

Response 6: By autoclaving, nest materials were under temperatures of more than 130 ºC and 15 psi of pressure for 30 minutes. We think we can safely assume that no insect pupae can survive at these conditions. The process described in lines 141-144 is the common process to separate Carnus from nest materials (Valera et al 2018 Parasitol Open), so we assume that we removed all pupae from the nest material. 

Point 7: Line 171: by means of.

Response 7: Corrected

Results section.

Point 8: In general; to my opinion this part is a bit weak and not balanced with the rest of the manuscript (other sections have far more text). I would suggest to say more about the data, also in relation to the next comment.

Response 8: Results are experimental and, thus, are more direct and easily described. Although we do not think the results section need of further explanation, we are open to do it if the reviewer indicates what part of the results should be further explained.

Point 9: On the order of data presentation: why not explain first figure 1 and then table 1? Or, put the table first in the manuscript. Readers would be inclined to look at the figure first. On this figure 1; the data presentation would benefit from minor changes: for consistency reasons, draw the uninterrupted line representing the trend for the closed circles and the interrupted line for the open circles. It is not logical this way.

Response 9a: Thanks. We have performed the requested changes and represented solid circles in association with continuous lines, while open circles were with discontinuous lines.

Second, the relevance of the connection in 1C is not clear to me. Why not show these data in bar graph form? Or please explain the line in the figure. The figure now suggests a relationship between different breeding seasons.

Response 9b: Ok, figure 1c is now a bar figure with no connecting lines.

Figure 1D: where in the figure can we see the ‘1’? I get the point, but maybe it is more logical to say that 0 is the start of the breeding season starting at the beginning of April.

Response 9c: We are not sure of catching the point. Breeding seasons start before first of April, but we give the value “1” to first of April. So, the zero in the X-axis is the 31st of March.

Point 10: About table 1: I would need more explanation. It is not clear to me for instance if the Mean numbers represent experimental versus controls or if they represent 2017 versus 2018 situations. I am kind of confused by the fact that ‘year’ is also mentioned as a variable on the left side in the table. Is there a way to make this information clearer?

Response 10a: the first lines of each model tested show mean values of experimental (Mean 1) and control nests (Mean 2), while the second lines show mean values of the 2017 (mean 1) and 2018 (mean 2) study years. We have clarified this in the table legend.

Minor comment: using comma and dots in the data table does not look very consistent.

Response 10b: Commas were used to separated degrees of freedom. We have added one blank space after each comma to avoid possible misleading interpretations.

Point 11: Minor comment: table 2 is referred to as table S1. And, it contains more or less the same information with additional information. Would it not be logical to show only new information or merge both tables?

Response 11: Following the suggestion by referee 1 we have removed the table S1.

Discussion.

Point 12: The discussion would benefit from opening with a general remark on the work which introduces the topics to be discussed.

Response 12a: we prefer to start with a summary of the results. The topic of the research is implicit in the las sentence when we suggested a general interpretation of the results.

The first sentence, line 246 is a bit awkward. Use words like ‘the treatment affected the intensity (or the number) of’. I understand that the presence of Carnus may influence results and that this issue was sorted out in the current study.

Response 12b: We rewrite the first sentence New lines 237-238“Our main results show that autoclaving nest material before reproduction reduced the intensity of ecto-parasitims of nestlings”.

We added a new test to discuss an alternative hypothesis to explain the possible influence of the presence of Carnus in nest-boxes.

New lines 315-322: “Another alternative hypothesis explaining our results is related to the possibility that C. hemapterus follows chemical cues from flies already present in hoopoe nests. However, for one of the study years, we removed C. hemapterus pupae from all collected nest-material and nest-boxes were newly installed, and hence, none of the parasitizing flies in experimental or control nests came from nest material that study year. Thus, because all parasitizing flies in our nest boxes should have come from other nests, chemicals from previous C. hemapterus activity would hardly explain the detected experimental effects in ecto-parasitim intensity.”.

To make lines 305 a bit more clear (please, have a look at the English here again), make a clear discrimination between what was done in which year.

Response 12c: rewritten:

New lines 201-303: “However, these results do not necessarily have to be the consequence of bacteria in the experimental nest-material, but rather of the emergence of adult ecto-parasites from overwintering pupae in the used nest material.”.

Since the studies described were done in 2017 and 2018, I assume that you are talking about non-published work in 2017 which is contrasting the results from the current study in which Carnus pupae were removed. This might be stated more clearly. If the above is not the case, then write in the materials and methods that experimentation was done different in 2017 compared with 2018.

Response 12c: the study was carried out in 2017 (with not pupae removed) and in 2018 (with pupae removed. That was explained in the Material and Method section (under the subheading 2.2 (Experimental design and fieldwork)).

Point 13: Lines 338 and 339: Experimental modification is necessary. How do you suggest to do this? And also, can you make suggestions on the methodology of profiling mentioned in lines 343 and further?

Response 13: It could be done by for instance injecting antibiotics into the uropygial gland of nestlings. The characterization of the volatiles presents in the nest and the hoopoe´s secretion could be conducted through the use of both chromatography-mass spectrometry. Regarding the nest volatile profile, the SPME fiber used in this technique can be placed directly in the nest box and subsequently analyzed. Finally, in the case of the volatile profile of secretion, it would be extracted into vials for further analysis. We have already done such kinds of samplings and are waiting for the resulting chemical profiles. Thus we know of the viability of these approaches.

Minor comments:

Point 14: Quite some typographical mistakes in the different sections.

Response 14: We have carefully reviewed the manuscript.

Round 2

Reviewer 1 Report

The text was shorten and improved. 

I am not an expert in English, but it is still difficult to follow the text in some cases. For example lines 30-31 can be modified as suggested:

30 These results suggest, on the one hand, a link between the community of microorganisms
31 of nest-material remains before reproduction started and intensity of ecto-parasitism, and, on the
32 other hand, that nest bacterial environment during reproduction predicts is related to fledging success.

Line 52

Instead of [12–14] [15–18] should be [12–18].

Line 77

Instead of  ….artificial hole-nests used by hoopoes (Upupa epops…. Should be ….artificial hole-nests used by hoopoes Upupa epops.

Line 87

Instead of   …the intensity of ecto-parasitation by Carnus hemapterus (hereafter C. hemapterus) flies, should be ….…the intensity of ecto-parasitation by Carnus hemapterus (hereafter C. hemapterus) flies.

I think abbreviation C. hemapterus is according to zoological nomenclature and is commonly used.

Table 1. It is not explained what does the bold numbers mean.

Line 248

Instead of   …volatiles associated to experimental hoopoe nests, several results, including the detected…

I would suggest …volatiles associated to experimental hoopoe nests, several our results, including the detected…

Line 251

…used by ecto-parasites should be mediated by symbiotic bacterial metabolism.

I doubt the word “symbiotic” in this case, because according to the results it seems that bacterial load  was positively associated with ecto-parasitism. Maybe these are symbionts of ecto-parasites?

Line 258-260

I would delete the sentence (starting from Unfortunately.....), because it does not provide new information. The text without this sentence seems not to be worse.

Author Response

Response to Reviewer 1 Comments

The text was shorten and improved. 

Point 1: I am not an expert in English, but it is still difficult to follow the text in some cases. For example lines 30-31 can be modified as suggested:

30 These results suggest, on the one hand, a link between the community of microorganisms
31 of nest-material remains before reproduction started and intensity of ecto-parasitism, and, on the
32 other hand, that nest bacterial environment during reproduction predicts is related to fledging success.

 Response 1: We have changed the sentence following your recommendation.

Point 2: Line 52

Instead of [12–14] [15–18] should be [12–18].

Response 2: Corrected

Point 3: Line 77

Instead of  ….artificial hole-nests used by hoopoes (Upupa epops…. Should be ….artificial hole-nests used by hoopoes Upupa epops.

 Response 3: Done

Point 4: Line 87

Instead of   …the intensity of ecto-parasitation by Carnus hemapterus (hereafter C. hemapterus) flies, should be ….…the intensity of ecto-parasitation by Carnus hemapterus (hereafter C. hemapterus) flies.

I think abbreviation C. hemapterus is according to zoological nomenclature and is commonly used.

Response 4: We have removed “(hereafter C. hemapterus)”

Point 5: Table 1. It is not explained what does the bold numbers mean.

Response 5: We have added in table 1 in line 225 “Statistical effects with associated alpha-values lower than 0.1 are highlighted in bold fonts”.

Point 6: Line 248

Instead of   …volatiles associated to experimental hoopoe nests, several results, including the detected…

I would suggest …volatiles associated to experimental hoopoe nests, several our results, including the detected…

Response 6: Changed

Point 7: Line 251

…used by ecto-parasites should be mediated by symbiotic bacterial metabolism.

I doubt the word “symbiotic” in this case, because according to the results it seems that bacterial load  was positively associated with ecto-parasitism. Maybe these are symbionts of ecto-parasites?

Response 7: The word “symbiotic” in this case is confusing so we have removed it.

Point 8: Line 258-260

I would delete the sentence (starting from Unfortunately.....), because it does not provide new information. The text without this sentence seems not to be worse.

Response 8: Following the reviewer advice, we have removed the sentence (“Unfortunately, we do not have information of the volatile profile of the experimental nest-boxes, nor of the uropygial secretion of females to explore possible experimental effects”).